# Recombinant Human Annexin A5 Ameliorates Localized Scleroderma by Inhibiting the Activation of Fibroblasts and Macrophages

**DOI:** 10.3390/pharmaceutics17080986

**Published:** 2025-07-30

**Authors:** Bijun Kang, Zhuoxuan Jia, Wei Li, Wenjie Zhang

**Affiliations:** Department of Plastic and Reconstructive Surgery, Shanghai 9th People’s Hospital, School of Medicine, Shanghai Jiao Tong University, Shanghai Key Laboratory of Tissue Engineering, National Tissue Engineering Center of China, 639 ZhiZaoJu Road, Shanghai 200011, China; kang_bj@foxmail.com (B.K.); 17853138537@163.com (Z.J.)

**Keywords:** annexin A5, localized scleroderma, anti-inflammation, anti-fibrosis, Smad2 phosphorylation

## Abstract

**Background**: Localized scleroderma (LoS) is a chronic autoimmune condition marked by cutaneous fibrosis and persistent inflammation. Modulating the activation of inflammatory cells and fibroblasts remains a central strategy in LoS treatment. We investigate the anti-fibrotic effects of Annexin A5 (AnxA5), identified as a key inflammatory component in fat extract, and assess its therapeutic efficacy. **Methods**: In vitro experiments were performed using TGF-β-stimulated primary human dermal fibroblasts treated with recombinant AnxA5. The anti-fibrotic effects and underlying mechanisms were assessed using CCK-8 assays, quantitative real-time PCR, Western blotting, and immunocytochemistry. In vivo, AnxA5 was administered via both preventative and therapeutic protocols in bleomycin-induced LoS mouse models. Treatment outcomes were evaluated by histological staining, collagen quantification, immunostaining, and measurement of pro-inflammatory cytokines. **Results**: TGF-β stimulation induced myofibroblast differentiation and extracellular matrix (ECM) production in dermal fibroblasts, both of which were significantly attenuated by AnxA5 treatment through the inhibition of phosphorylation of Smad2. In vivo, both preventative and therapeutic administration of AnxA5 effectively reduced dermal thickness, collagen deposition, ECM accumulation, M1 macrophage infiltration, and levels of pro-inflammatory cytokines. **Conclusions**: Through both preventative and therapeutic administration, AnxA5 ameliorates LoS by exerting dual anti-fibrotic and anti-inflammatory effects, underscoring its potential for treating fibrotic diseases.

## 1. Introduction

Localized scleroderma (LoS) is a rare and complex autoimmune connective tissue disorder characterized by cutaneous fibrosis and chronic inflammation, with an estimated prevalence of fewer than six cases per 100,000 individuals and substantial associated morbidity [1,2]. The pathogenesis of LoS involves vasculopathy and dysregulation of both innate and adaptive immune responses, ultimately leading to excessive inflammatory cell infiltration and aberrant extracellular matrix (ECM) deposition [3]. Clinically, LoS progresses through three distinct phases: the edematous, fibrotic, and atrophic stages [4]. Current therapeutic strategies primarily rely on immunosuppressive agents such as glucocorticoids and methotrexate; however, their clinical efficacy is limited, and long-term use is associated with significant adverse effects [5]. Fat grafting has emerged as a promising alternative therapy for LoS, offering multiple benefits including anti-inflammatory activity, attenuation of pathological ECM accumulation, and restoration of soft tissue volume [6]. Clinical studies have reported an efficacy rate of 87.3% for fat grafting in LoS patients, highlighting its substantial potential to improve clinical outcomes [7].

The therapeutic effects of fat grafting are mediated by its cellular components, including adipocytes, stromal vascular fraction (SVF), and adipose-derived mesenchymal stem cells (ADSCs). Dedifferentiated and redifferentiated adipocytes contribute to subcutaneous fat regeneration and reversal of skin fibrosis [8]. The SVF contains a heterogeneous population of cells that secrete paracrine factors with pro-angiogenic and anti-inflammatory properties [9]. In addition, ADSCs have demonstrated strong anti-fibrotic and immunosuppressive effects in models of skin fibrosis, evidenced by reduced immune cell infiltration and downregulation of fibrogenic markers [10]. Our previous work identified cell-free fat extract (CEFFE), derived from adipose tissue, as a protein-rich product capable of modulating inflammation and promoting tissue repair [11,12,13]. Studies in multiple animal models and clinical trials suggest that CEFFE mimics the paracrine activity of stem cells [14,15,16,17,18]. For example, clinical evaluation of CEFFE in post-inflammatory hyperpigmentation secondary to radiation exposure demonstrated improved skin texture, supporting its anti-inflammatory and anti-fibrotic properties [19]. However, CEFFE is a complex liquid mixture containing more than 1700 protein components, which poses a challenge for therapeutic standardization. To address this, we employed protein fractionation and bioactivity-guided screening to identify annexin A5 (AnxA5) as a key anti-inflammatory component of CEFFE [20].

Annexin A5 (AnxA5) is a calcium-dependent phospholipid-binding protein that interacts with negatively charged phospholipids on the cell membrane [21,22]. In various inflammatory disease models, AnxA5 has been shown to play a key role in regulating macrophage-mediated immune responses, underscoring its potent anti-inflammatory properties [23,24,25]. Specifically, AnxA5 suppresses M1 macrophage polarization while promoting M2 polarization. It has also been reported to downregulate the expression of α-smooth muscle actin (α-SMA) and transforming growth factor-beta (TGF-β), thereby reducing collagen deposition and mitigating the progression of hepatic fibrosis in models of nonalcoholic steatohepatitis [23]. Although these findings clarify the anti-inflammatory mechanisms of AnxA5, its potential anti-fibrotic effects and associated molecular pathways remain to be fully elucidated.

In fibrotic skin lesions, the infiltration of immune cells triggers the release of inflammatory mediators such as TGF-β, platelet-derived growth factor (PDGF), and interleukin-13 (IL-13), which act in a paracrine manner on resident skin cells [26]. Fibroblasts, the key effector cells in fibrotic processes, become activated and differentiate into myofibroblasts, leading to the excessive synthesis and secretion of extracellular matrix (ECM) components, particularly type I collagen (COL I), type III collagen (COL III), and fibronectin (FN) [27]. The TGF-β/Smad signaling pathway plays a central role in regulating fibrogenesis [28]. Upon binding of TGF-β ligands to their cognate receptors, intracellular signaling is initiated through the phosphorylation of Smad2 and Smad3, which then translocate into the nucleus to activate the transcription of pro-fibrotic genes [29]. Given its pivotal role in driving fibroblast activation and ECM production, targeted modulation of the TGF-β/Smad pathway has emerged as a key therapeutic approach in the treatment of fibrotic diseases. In this study, we investigated the anti-fibrotic effects of AnxA5 and its regulatory role in TGF-β/Smad signaling in dermal fibroblasts. Our results demonstrated that AnxA5 suppresses TGF-β-induced phosphorylation and nuclear translocation of Smad2, thereby inhibiting fibroblast activation and extracellular matrix secretion. To model localized scleroderma (LoS), mice were subjected to repeated subcutaneous bleomycin injections [30]. Using both preventative and therapeutic treatment strategies, we systematically evaluated the efficacy of AnxA5 in modulating fibrotic progression and inflammatory responses in bleomycin-induced LoS murine models.

## 2. Methods and Materials

### 2.1. Cell Culture

Primary human dermal fibroblasts (HDFs) were isolated from skin tissue samples obtained from five pediatric patients (aged 6–10 years) undergoing routine circumcision at Shanghai 9th People’s Hospital, with informed consent obtained from all patients and their guardians. The dermal fibroblasts were isolated through enzymatic digestion using neutral protease and collagenase, and subsequently cultured in Dulbecco’s Modified Eagle Medium (DMEM) supplemented with 10% fetal bovine serum, 100 U/mL penicillin, and 100 μg/mL streptomycin at 37 °C in a humidified incubator with 5% CO_2_. Cells at passages 1 to 3 were used for all experiments. Culture medium was refreshed every two days, and cells were used once they reached 80–90% confluency after passaging. For in vitro assays, HDFs were treated with recombinant human AnxA5 (0, 2.5, 5, or 10 μg/mL; SEME Cell Technology Co., Ltd., Shanghai, China) and the simultaneous treatment of 10 ng/mL recombinant human TGF-β (PeproTech, Rocky Hill, NJ, USA). The human immortalized keratinocyte cell line HaCaT (ATCC, Manassas, VA, USA) was cultured under the same conditions as HDFs. HaCaT cells were treated with recombinant human AnxA5 (0, 2.5, 5, or 10 μg/mL) in the presence or absence of a pro-inflammatory cytokine cocktail composed of IL-17A, IL-22, and TNF-α (each at 10 ng/mL; PeproTech, Rocky Hill, NJ, USA). All cell culture experiments were conducted at 37 °C in a humidified 5% CO_2_ atmosphere, with medium changes every 48 h.

### 2.2. Cell Counting Kit 8 (CCK-8) Assay

For cell proliferation assays, HDFs or HaCaT cells were seeded in 96-well culture plates at a density of 3 × 10^3^ cells per well and allowed to adhere for 24 h under standard culture conditions. Following treatments, cellular proliferation was assessed at 24, 48, and 72-h time points using the CCK-8 kit (Dojindo Molecular Technologies, Kumamoto, Japan). The kit quantifies the number of live cells through a water-soluble tetrazolium salt, producing an orange formazan. Briefly, culture medium was aspirated and replaced with 100 μL of 1:10 diluted CCK-8 reagent in serum-free medium. Plates were subsequently incubated at 37 °C for 2 h in a humidified 5% CO_2_ atmosphere. Absorbance was measured at 450 nm using a SpectraMax^®^ i3x microplate reader (Molecular Devices, San Jose, CA, USA). Relative optical density (O.D.) values were calculated using the following formula: Relative O.D. = (O.D. experimental group) − (O.D. blank control).

### 2.3. Quantitative Real-Time PCR

Total RNA was extracted from cultured cells using the RNA Purification Kit (B0004DP; EZ Bioscience, College Park, MD, USA) according to the manufacturer’s instructions. RNA concentration and purity were determined spectrophotometrically. Subsequently, 1 μg of total RNA was reverse transcribed into complementary DNA (cDNA) using the 4× Reverse Transcription Master Mix (A0010CGQ; EZBioscience^®^). Quantitative real-time PCR (qPCR) was performed using SYBR™ Green qPCR Master Mix (A0012-R2; EZBioscience^®^) and gene-specific primers (sequences provided in Appendix A) on a real-time PCR detection system. All reactions were conducted in technical triplicates. Relative gene expression levels were normalized to GAPDH as an internal control and calculated using the 2^−ΔΔCt^ method, with results expressed as fold-changes relative to control groups.

### 2.4. Western Blotting

HDFs were seeded into six-well plates at a density of 1 × 10^5^ cells per well and allowed to adhere overnight. Protein lysates were collected 72 h post-treatment and subjected to sodium dodecyl sulfate-polyacrylamide gel electrophoresis (SDS-PAGE), followed by membrane transfer and blocking. Membranes were incubated with primary antibodies overnight at 4 °C and then with secondary antibodies for 1 h at room temperature prior to visualization. The primary antibodies used included: anti-COL I antibody (1:1000; 14695-1-AP; Proteintech, Wuhan, China), anti-α-SMA antibody (1:1000; ab5694; Abcam, Cambridge, UK), anti-p-Smad2 antibody (1:1000; 18338S; CST, Danvers, MA, USA), anti-p-Smad3 antibody (1:1000; 9520S; CST), and anti-Smad2/3 antibody (1:1000; 8685S; CST). The secondary antibody used was a horseradish peroxidase-conjugated rabbit antibody (1:10,000; SA00001-2; Proteintech, Wuhan, China). Protein bands were acquired and quantified using iBright Imager and iBright Analysis Software 5.4.0 (Thermo Fisher Scientific, Waltham, MA, USA), with results (Local Bg. Corr. Vol.) normalized to GAPDH and presented as a ratio relative to the control group.

### 2.5. Immunofluorescence Staining

Immunofluorescence staining was performed on cell coverslips and skin tissue sections following standard immunofluorescence procedures. Samples were incubated with primary antibodies overnight at 4 °C, followed by incubation with fluorescently labeled secondary antibodies for 1 h at room temperature. Cell nuclei were counterstained with DAPI (P0131; Beyotime, Shanghai, China). The primary antibodies used included: anti-α-SMA antibody (1:200; ab7817; Abcam, Cambridge, UK), anti-PCNA antibody (1:500; ab29; Abcam, Cambridge, UK), anti-CD68 antibody (1:100; ab303565; Abcam, Cambridge, UK), anti-CD86 antibody (1:200; 19589S; CST), anti-COL I antibody (1:200; 14695-1-AP; Proteintech, Wuhan, China), and anti-p-Smad2 antibody (1:200; 18338; CST). The secondary antibodies were Alexa Fluor^®^ 488-conjugated mouse anti-rabbit (1:1000; CST) and Alexa Fluor^®^ 594-conjugated rabbit anti-mouse (1:1000; CST). Images were acquired using a confocal fluorescence microscope (Leica, Wetzlar, Germany) equipped with the Leica LAS X system. Quantitative analysis of fluorescence intensity (/mm^2^) or positive cell counts was performed using ImageJ software V.1.8.0 (NIH). Three sections were analyzed for each sample.

### 2.6. Establishment and Treatment in Los Murine Models

All animal procedures were approved by the Animal Care and Experiment Committee of Shanghai Jiao Tong University School of conducted in accordance with the National Research Council’s Guide for the Care and Use of Laboratory Animals. Male C57BL/6 mice (6–8 weeks old, 23.48 ± 1.85 g) were obtained from Vital River Laboratories (Beijing, China) and housed under specific pathogen-free conditions. Mice were provided ad libitum access to food and water and maintained under controlled environmental conditions: temperature (22 ± 1 °C), relative humidity (55 ± 5%), and a 12 h light/dark cycle. After a 7-day acclimatization period, mice were anesthetized via intraperitoneal injection of pentobarbital sodium (50 mg/kg). A 1 cm^2^ area on the dorsal skin was shaved and marked to define the injection site. To establish the localized scleroderma (LoS) model, mice were randomly divided into two groups. The control group (CTRL, *n* = 3) receieved subcutaneous injections of 100 μL sterile normal saline (NS, 0.9% NaCl) while the model group (BLM, *n* = 3) received subcutaneous injections of 100 μL bleomycin solution (0.5 mg/mL; Nippon Kayaku Co., Ltd., Tokyo, Japan) into the marked dorsal area every other day. Dorsal skin samples were harvested at week 3 post-injection for validation of model establishment.

To evaluate the therapeutic efficacy of systemic AnxA5 administration, mice were randomly assigned to four experimental groups (*n* = 6 per group). Group 1 (CTRL) received subcutaneous injections of 100 μL normal saline (NS) every other day. Group 2 (BLM) received subcutaneous injections of 100 μL bleomycin solution (0.5 mg/mL) and intraperitoneal injections of 200 μL NS every other day. Group 3 (AnxA5-6w) received subcutaneous injections of 100 μL bleomycin solution (0.5 mg/mL) and intraperitoneal injections of 200 μL AnxA5 solution (0.2 mg/mL) every other day for 6 weeks. Group 4 (AnxA5-3w) received subcutaneous bleomycin injections (100 μL, 0.5 mg/mL) every other day for 6 weeks and intraperitoneal AnxA5 injections (200 μL, 0.2 mg/mL) every other day from weeks 4 to 6. The dosage of AnxA5 was based on prior studies in osteoarthritis models [20] and aligned with protocols used in other bleomycin-induced skin fibrosis models [31]. At the end of week 6, dorsal skin samples were collected from all groups for further analysis.

For topical AnxA5 application studies, mice were randomly assigned to four groups (*n* = 6 per group): Group 1 (CTRL) received subcutaneous injections of 100 μL normal saline (NS) every other day. Group 2 (BLM) received subcutaneous injections of 100 μL bleomycin solution (0.5 mg/mL) along with subcutaneous injections of 200 μL NS every other day. Group 3 (AnxA5-6w) received subcutaneous injections of 100 μL bleomycin solution (0.5 mg/mL) and topical subcutaneous injections of 10 μL AnxA5 solution (2 mg/mL) every other day for 6 weeks. Group 4 (AnxA5-3w) received subcutaneous bleomycin injections (100 μL, 0.5 mg/mL) every other day for 6 weeks, and topical subcutaneous injections of 10 μL AnxA5 solution (2 mg/mL) every other day from week 4 to week 6. At the end of the sixth week, dorsal skin samples were harvested from all groups for further evaluation.

### 2.7. Histological Staining

Following tissue collection, skin samples were fixed in 4% paraformaldehyde (PFA), processed through standard dehydration procedures, and embedded in paraffin. Paraffin blocks were sectioned into 5 μm thick slices using a microtome. Tissue sections were then subjected to hematoxylin and eosin (H&E), Masson’s trichrome, and Sirius red staining according to the manufacturer’s protocols. Sections were sequentially dewaxed, rehydrated, and stained with hematoxylin for 10 min. For H&E staining, sections were counterstained with eosin for 3 min. For Masson’s trichrome staining, sections were blued in Masson bluing solution for 3 min, stained with Ponceau-fuchsin solution for 5 min, differentiated in phosphomolybdic acid solution for 1 min, and stained with aniline blue for 2 min. For Sirius red staining, sections were incubated in Sirius red solution for 30 min. All stained sections were mounted with neutral resin and imaged using an optical microscope (Nikon Instruments, Tokyo, Japan).

### 2.8. Collagen Content Detection

Skin tissues were hydrolyzed in lysis buffer, and the pH of the resulting lysates was adjusted to 6–8 using sodium hydroxide (NaOH). Hydroxyproline content was quantified using a commercial hydroxyproline assay kit (Nanjing Jiancheng Bioengineering Institute, Nanjing, China) according to the manufacturer’s protocol. Briefly, 100 μL of each sample was mixed with assay reagents and incubated at 60 °C for 15 min. Absorbance was measured at 550 nm using a microplate reader. A standard curve was generated using hydroxyproline standards ranging from 0 to 200 μg/mL. Hydroxyproline content was calculated using the following formula:Hydroxyproline content (μg/mg tissue) = (Sample O.D. − Blank O.D.)/
(Slope of standard curve) × Dilution factor/Tissue weight (mg)

All samples were measured in triplicate, and results were normalized to tissue wet weight. Hydroxyproline content was used as an indicator of total collagen content, with a conversion factor of 7.46 (1 μg hydroxyproline ≈ 7.46 μg collagen).

### 2.9. Immunohistochemical Staining

Skin tissue samples were fixed in 4% PFA for 24 h at 4 °C, followed by standard tissue processing through graded ethanol dehydration, xylene clearing, and paraffin embedding. The pre-cooled paraffin blocks were sectioned into slices with a thickness of 5 μm. Sections were dewaxed, rehydrated, and subjected to antigen retrieval by immersion in citrate buffer, followed by microwave heating at high power for 2 min and incubation in a 98 °C water bath for 30 min. Permeabilization was performed by soaking the sections in 0.2% Triton X-100 solution for 5 min. Endogenous peroxidase activity was blocked using hydrogen peroxide for 10 min. Sections were then incubated in 5% goat serum for 30 min to block non-specific binding sites. Primary antibodies anti-fibronectin (FN) (1:600; 15613-AP; Proteintech, Wuhan, China) were applied and incubated overnight at 4 °C. The next day, HRP-conjugated secondary antibodies (rabbit anti-mouse, 1:1000; goat anti-rabbit, 1:1000) were applied for 30 min at room temperature. Color development was achieved using 3,3′-diaminobenzidine (DAB) solution, and the reaction was stopped at the appropriate endpoint. Sections were mounted with neutral resin and visualized under an optical microscope. Quantitative analysis of immunohistochemical staining was performed using ImageJ software.

### 2.10. Enzyme-Linked Immunosorbent Assay (ELISA)

Skin tissue samples were homogenized in ice-cold RIPA buffer supplemented with protease inhibitors using a tissue homogenizer. The homogenates were centrifuged at 12,000× *g* for 15 min at 4 °C, and the resulting supernatants were collected for further analysis. Total protein concentration was determined using a bicinchoninic acid (BCA) assay kit (Beyotime Biotechnology, Shanghai, China). Cytokine levels were measured using commercial ELISA kits (Elabscience Biotechnology, Houston, TX, USA) according to the manufacturer’s protocols. Briefly, 100 μL of tissue lysates or standards were added to antibody-precoated 96-well plates and incubated at 37 °C for 2 h. Following washes, biotinylated detection antibodies were added and incubated for 1 h at 37 °C, followed by incubation with streptavidin-HRP conjugate for 30 min. Signal development was performed using 3,3′,5,5′-tetramethylbenzidine (TMB) substrate and stopped with 2 N sulfuric acid. Absorbance was measured at 450 nm using a microplate reader. The concentrations of TNF-α, IL-6, IL-17A, and IL-22 were calculated from standard curves and normalized to total protein content, expressed as pg cytokine/mg total protein. All samples were analyzed in duplicate to ensure reproducibility.

### 2.11. Statistical Analysis

All quantitative data are presented as mean ± standard error of the mean (SEM) from at least three independent experiments. Statistical analyses were conducted using GraphPad Prism version 9.3 (GraphPad Software, San Diego, CA, USA). Comparisons between two groups were performed using unpaired two-tailed Student’s *t*-tests. For comparisons among multiple groups, one-way analysis of variance (ANOVA) followed by Tukey’s post hoc test was applied. Statistical significance was defined as *p* < 0.05, with the following notation used to indicate significance levels: * *p* < 0.05, ** *p* < 0.01, *** *p* < 0.001, and **** *p* < 0.0001.

## 3. Results

### 3.1. Effect of AnxA5 on TGF-β-Induced Dermal Fibroblasts

HDFs are recognized as pivotal effector cells in fibrotic processes [32]. To investigate the anti-fibrotic potential of AnxA5, we established an in vitro fibrotic model using TGF-β stimulation. As an initial observation, TGF-β treatment enhanced HDF proliferation, whereas AnxA5 did not alter this proliferative response (Figure 1A). qRT-PCR analysis demonstrated that TGF-β induced myofibroblast differentiation with the increased expressions of pro-fibrotic markers (ACTA2, CTGF, TGFB1), while treatment with AnxA5 significantly attenuated TGF-β-induced expression of these fibrotic markers (Figure 1B). TGF-β induced ECM synthesis and attenuated ECM degradation with the increased expressions of ECM-related genes (COL1A1, FN1) and the downregulation of matrix degradation-related genes (MMP1/TIMP1), while treatment with AnxA5 attenuated TGF-β-induced expression of these ECM-related genes and restored the MMP1/TIMP1 ratio toward normal levels (Figure 1C). At the protein level, Western blot analysis confirmed that AnxA5 treatment suppressed TGF-β-induced expressions of α-SMA and COL I (Figure 1D). Consistent with these findings, immunofluorescence analysis revealed that TGF-β stimulation led to increased cytoplasmic COLI expression, which was markedly reduced following AnxA5 treatment (Figure 1E). Quantitative analysis revealed a significant reduction in COLI fluorescence intensity in AnxA5-treated cells compared to TGF-β-stimulated controls (Figure 1E). Collectively, these findings demonstrate that while TGF-β-activated dermal fibroblasts undergo myofibroblast differentiation and ECM production, AnxA5 effectively attenuates these fibrotic responses without interfering with cellular proliferation, supporting its anti-fibrotic property.

### 3.2. AnxA5 Inhibited TGF-β-Induced Activation of the TGF-β/Smad2 Signaling Pathway

To elucidate the molecular mechanisms underlying AnxA5-mediated inhibition of TGF-β signaling in HDFs, we investigated its effects on the Smad2/3 signaling pathway. Western blot analysis revealed that TGF-β stimulation significantly increased the phosphorylation of both Smad2 and Smad3. Notably, AnxA5 treatment selectively inhibited TGF-β-induced phosphorylation of Smad2, with no significant effect on Smad3 phosphorylation (Figure 2A). Immunofluorescence staining further confirmed these findings, demonstrating that TGF-β stimulation induced nuclear translocation of phosphorylated Smad2 (p-Smad2), which was markedly reduced by AnxA5 treatment. Quantitative analysis revealed a significant reduction in p-Smad2 fluorescence intensity in AnxA5-treated cells compared to TGF-β-stimulated controls (Figure 2B). These results demonstrate that AnxA5 exerts its anti-fibrotic effects through selective inhibition of the TGF-β/Smad2 signaling pathway, thereby suppressing TGF-β-induced fibroblast activation.

### 3.3. AnxA5 Ameliorated Bleomycin-Induced LoS in Murine Models

To validate the LoS murine model, we systematically characterized bleomycin-induced skin pathology. Histological analysis using H&E and MASSON trichrome staining revealed significant dermal thickening and collagen deposition at both 3- and 6-week time points compared to the CTRL groups (Appendix AA–C and Figure 3B–F). Sirius red staining under polarized light microscopy showed a distinct shift in collagen composition in bleomycin-treated mice, with increased red-yellow birefringent fibers (indicative of mature COLI) and decreased yellow-green fibers (indicative of COLIII) (Appendix AA; Figure 4C). Quantification of hydroxyproline further supported these observations, revealing significantly elevated total collagen content in the BLM groups relative to controls (Appendix AD; Figure 3G). These results confirmed the successful establishment of a fibrotic LoS murine model.

Preventative treatment with AnxA5 for 6 weeks (AnxA5-6w group) significantly reduced both epidermal and dermal thickness and decreased the extent of collagen staining in Masson’s trichrome sections compared to the BLM group. In the therapeutic group (AnxA5-3w), which received AnxA5 from weeks 4 to 6 after fibrosis was established, epidermal thickness and collagen-stained areas were reduced, although dermal thickness remained largely unchanged (Figure 3B–F). Quantitative analysis of total collagen content confirmed a reduction in collagen levels in both AnxA5-6w and AnxA5-3w groups, with the most pronounced effect observed in the preventative treatment group (Figure 3G). These results demonstrate that AnxA5 effectively modulates collagen deposition and composition in bleomycin-induced LoS murine models, with greater therapeutic efficacy observed under preventative administration.

### 3.4. AnxA5 Inhibited Myofibroblast Activation and ECM Secretion

To investigate the regulatory role of AnxA5 in myofibroblast activation in LoS, immunofluorescence staining for α-SMA was performed on skin tissue sections. The results demonstrated a marked increase in α-SMA expression in the bleomycin-induced group (BLM) compared to the control group (CTRL). Following AnxA5 treatment, both the preventive treatment group (AnxA5-6w) and the therapeutic treatment group (AnxA5-3w) exhibited significantly reduced α-SMA expression relative to the BLM group, with the lowest expression observed in the AnxA5-6w group (Figure 4A,B). Sirius red staining further revealed a reduced COL I/COL III ratio in AnxA5-treated groups compared to the BLM group (Figure 4C,D). Consistently, immunohistochemical staining for FN indicated decreased FN expression in AnxA5-treated groups relative to the BLM group (Figure 4E,F). These findings collectively demonstrate that AnxA5 treatment suppresses myofibroblast activation, thereby reducing ECM secretion.

### 3.5. AnxA5 Modulated Inflammation in Bleomycin-Induced LoS

Excessive inflammatory responses are known to impede tissue regeneration and repair [33]. To investigate the immunomodulatory effects of AnxA5 in bleomycin-induced LoS, we characterized the inflammatory response and its modulation following AnxA5 treatment. Immunofluorescence staining for the macrophage marker CD68 revealed a significant increase in the number of macrophages infiltrated in the BLM group compared to the CTRL group. In contrast, both AnxA5-6w and AnxA5-3w groups exhibited a marked reduction in the macrophages compared to the BLM group (Figure 5A,B), suggesting that AnxA5 regulates macrophage-mediated inflammation during fibrotic progression. Further characterization of macrophage polarization demonstrated an increase in CD86+ M1-phenotype macrophages in bleomycin-treated skin compared to the controls, whereas AnxA5 treatment significantly reduced the presence of M1-phenotype macrophages in both treatment groups (Figure 5C,D). Pro-inflammatory cytokines such as TNF, IL-6, IL-17A, and IL-22 contribute to chronic inflammation and play key roles in promoting fibrosis [34,35]. ELISA-based quantification of these cytokines in skin tissue revealed elevated levels in the BLM group compared to the CTRL group. Treatment with AnxA5 significantly reduced the concentrations of TNF, IL-6, IL-17A, and IL-22 (Figure 5E), indicating effective suppression of the inflammatory response. Collectively, these results demonstrate that AnxA5 modulates macrophage-driven inflammation in bleomycin-induced LoS by reducing macrophage infiltration, inhibiting polarization toward the pro-inflammatory M1 phenotype, and attenuating the production of pro-inflammatory cytokines.

### 3.6. AnxA5 Inhibited Epidermal Hyperproliferation in Bleomycin-Induced LoS

Inflammatory cytokines released following tissue injury can stimulate epidermal cell proliferation, leading to epidermal thickening. Histological staining analysis confirmed that bleomycin-induced LoS led to pronounced epidermal thickening, which was alleviated by AnxA5 treatment (Figure 3B,C). To further investigate the impact of AnxA5 on epidermal proliferation, immunofluorescence staining for the proliferation marker PCNA was performed on skin tissue sections. The BLM group exhibited increased PCNA expression and a greater number of PCNA-positive cells in the epidermis compared to the CTRL group. In contrast, AnxA5 treatment significantly reduced PCNA expression and the number of proliferating epidermal cells in both the AnxA5-6w and AnxA5-3w groups (Figure 6A,B). To assess this effect in vitro, an inflammatory microenvironment was simulated by treating HaCaT keratinocytes with a cytokine cocktail consisting of TNF-α, IL-17A, and IL-22. This treatment promoted HaCaT proliferation, whereas AnxA5 treatment at concentrations of 2.5, 5, and 10 μg/mL effectively suppressed this cytokine-induced proliferation (Figure 6C). These findings indicate that AnxA5 attenuates inflammation-driven epidermal hyperproliferation, thereby contributing to the reduction in epidermal thickening observed in fibrotic skin lesions in LoS.

### 3.7. Topical Application of AnxA5 in Bleomycin-Induced Skin Fibrosis

Building upon the demonstrated efficacy of systemic AnxA5 administration, we further explored the therapeutic potential of topical AnxA5 application using both preventative and therapeutic approaches in the bleomycin-induced LoS model (Figure 7A). Histological analysis revealed that topical AnxA5 treatment significantly reduced epidermal thickness in both the preventative (AnxA5-6w) and therapeutic (AnxA5-3w) groups compared to the BLM control (Figure 7B,C). Dermal thickness was also attenuated with a notable reduction observed in the AnxA5-6w group (Figure 7B,D). Histopathological evaluation using Masson’s trichrome and Sirius red staining further demonstrated significant reductions in collagen deposition both AnxA5-treated groups (Figure 7B,E). These findings were supported by hydroxyproline quantification, which confirmed a reduction in total collagen content in both the AnxA5-6w and AnxA5-3w groups (Figure 7F). Collectively, these results suggested that the preventative administration of AnxA5 yields more pronounced anti-fibrotic effects than therapeutic intervention during the later stages of disease progression. Furthermore, ELISA-based analysis of pro-inflammatory cytokines revealed that topical AnxA5 treatment significantly reduced bleomycin-induced elevations of TNF-α, IL-6, IL-17A, and IL-22 (Figure 7G). These outcomes closely mirrored the results observed with systemic AnxA5 delivery, supporting the robustness of AnxA5’s anti-inflammatory and anti-fibrotic effects across different modes of administration. Taken together, these findings highlight topical AnxA5 as a promising and clinically relevant therapeutic strategy for LoS, with preventative application offering superior efficacy compared to delayed intervention.

## 4. Discussion

Localized scleroderma (LoS), though rare, has a profound impact on skin tissue structure and function, particularly when affecting the face. Effective treatment strategies for LoS focus on controlling both fibrotic progression and inflammation. In this study, we provide the first evidence that AnxA5 possesses anti-fibrotic activity by regulating extracellular matrix (ECM) synthesis and degradation through inhibition of the TGF-β/Smad2 signaling pathway in dermal fibroblasts. In bleomycin-induced murine models, systemic application of AnxA5 alleviated LoS pathology by modulating both fibroblasts and macrophages activity. Specifically, AnxA5 suppressed the expression of α-SMA, a myofibroblast marker, thereby reducing collagen synthesis and secretion. Concurrently, AnxA5 downregulated the expressions of markers of pro-inflammatory M1 macrophages (CD68 and CD86), leading to decreased production of inflammatory cytokines. These findings suggest that AnxA5, as a single-molecule therapeutic agent, exerts dual anti-fibrotic and immunomodulatory effects. Through both preventative and therapeutic administration, AnxA5 shows promising potential to improve outcomes in LoS and other fibrotic skin diseases.

LoS is characterized by persistent inflammation and progressive fibrosis involving both the dermis and subcutaneous adipose tissue. As the disease progresses, excessive extracellular matrix (ECM) deposition invades and replaces adipose tissue, ultimately compromising the structural and functional integrity of the skin [36]. Previous studies have reported that AnxA5 is abundantly expressed in adipose tissue but is significantly downregulated in the skin [37], suggesting a potential deficiency of endogenous AnxA5 in LoS-affected skin lesions. In this study, we demonstrated that exogenous supplementation of recombinant AnxA5 effectively attenuated both fibrosis and inflammation in LoS murine models. This finding is consistent with previous reports that AnxA5 attenuates hepatic fibrosis and ECM deposition in high-fat diet-induced nonalcoholic steatohepatitis by downregulating α-SMA, COL III, and TGF-β expression [23]. We further validated the anti-fibrotic properties of AnxA5 in dermal fibroblasts (Figure 1 and Figure 2). TGF-β signaling includes both canonical (Smad-dependent) and non-canonical (JNK, ERK, and PI3K/AKT) pathways. While these pathways exhibit cross-talk with Smad signaling and ultimately converge on nuclear Smad complexes, the canonical TGF-β/Smad pathway remains the primary mediator of fibrogenesis [38]. In this study, we specifically detected Smad2/3 activation following AnxA5 treatment. Under TGF-β stimulation, AnxA5 significantly inhibited Smad2 phosphorylation and its nuclear translocation, thereby disrupting the canonical TGF-β/Smad2 signaling pathway (Figure 2). As a result, AnxA5 suppressed the expression of pro-fibrotic markers including α-SMA, CTGF, and TGF-β, preventing fibroblast activation and myofibroblast differentiation. In addition, AnxA5 reduced the synthesis and secretion of ECM components such as COL I and FN, while promoting ECM degradation by increasing the MMP1/TIMP1 ratio (Figure 1). While our data demonstrate AnxA5’s selective inhibition of canonical TGF-β/Smad2 signaling (Figure 2), we cannot exclude potential modulation of non-canonical pathways that may synergistically contribute to its anti-fibrotic effects. Future studies could systematically evaluate AnxA5’s broader signaling regulation while maintaining therapeutic specificity. These findings uncover a novel biological function of AnxA5 as a potent anti-fibrotic regulator in dermal fibroblasts and highlight its therapeutic potential for the treatment of fibrotic skin disorders such as LoS.

LoS is generally recognized as a biphasic disease, consisting of an initial inflammatory stage followed by a fibrotic phase [39]. The early phase is characterized by aberrant immune activation and heightened inflammatory responses, with prominent infiltration of immune cells and increased secretion of pro-inflammatory cytokines [40]. Current therapeutic strategies emphasize the importance of early immunomodulatory intervention to impede disease progression and prevent late-stage fibrosis [41]. In this study, we evaluated the therapeutic efficacy of AnxA5 in bleomycin-induced LoS models using both preventative and regressive treatment approaches. Comparative analysis revealed that preventative treatment (AnxA5-6w group) demonstrated superior immunomodulatory effects, as evidenced by a greater reduction in CD86+ M1 macrophage infiltration and more pronounced suppression of pro-inflammatory cytokine production compared to the regressive treatment group (AnxA5-3w) (Figure 6). In terms of fibrotic outcomes, the AnxA5-6w group also showed enhanced therapeutic benefit, including greater reductions in epidermal and dermal thickness, decreased collagen content, and attenuated ECM deposition (Figure 1 and Figure 2). These results suggest that early intervention with AnxA5 more effectively inhibits dermal fibroblast activation and myofibroblast differentiation, while also modulating M1 macrophage-dominated inflammation. While the current study establishes AnxA5’s capacity to reduce macrophage infiltration and pro-inflammatory cytokine secretion in LoS lesions at the phenotypic level, detailed mechanistic insights into AnxA5-mediated macrophage polarization are systematically characterized in our other studies [20,42]. The superior efficacy observed with preventative treatment highlights the critical importance of early therapeutic intervention in LoS and supports the potential of AnxA5 as a preventative strategy to halt disease progression.

In addition to systemic administration, we evaluated the therapeutic potential of topical AnxA5 application in bleomycin-induced LoS models. This localized delivery strategy offers distinct pharmacological advantages, including enhanced drug concentration at the lesion site and reduced risk of systemic adverse effects [43]. Multiple dermal delivery systems exist for recombinant proteins. Due to their large molecular weight and poor biological membrane permeability, techniques such as fractional laser ablation or microneedle rollers can enhance transdermal delivery by creating micropore arrays [44,45]. Liposome-encapsulated recombinant human growth hormone modified with hyaluronic acid and incorporated into carbomer gel significantly increased the secretion of COL I and ameliorated skin photoaging [46]. Similarly, microneedle-mediated delivery of recombinant human epidermal growth factor accelerated wound healing [47]. In this study, recombinant human AnxA5 was administered via subcutaneous injection to treat localized scleroderma. Our findings demonstrated that topical AnxA5 administration significantly reduced collagen deposition and suppressed cutaneous inflammation, effectively ameliorating LoS under both preventative and therapeutic administration regimens (Figure 7). Collectively, our study provides comprehensive evidence supporting the efficacy of AnxA5 in LoS management through both systemic and topical delivery strategies. In the bleomycin-induced LoS model, both approaches exhibited robust anti-inflammatory and anti-fibrotic effects. While recombinant protein therapeutics offer significant potential, challenges remain regarding limited storage stability and potential immunogenicity. Recent advances in protein engineering, including chemical modifications and innovative transdermal delivery systems, have shown promise in mitigating these limitations while maintaining therapeutic efficacy [48]. Nevertheless, further research is warranted to optimize therapeutic parameters such as administration route, dosage, and delivery formulation to maximize efficacy and support clinical translation. In particular, the development of AnxA5-based topical formulations with improved skin penetration and sustained release properties represents a promising avenue for future investigation.

AnxA5 has been extensively studied for its diverse biological functions, including anti-coagulant, anti-inflammatory, and anti-apoptotic activities [49], positioning it as a novel therapeutic candidate for inflammation-associated disorders. The clinical translation of recombinant human AnxA5 is supported by completed phase I trials (NCT04217629, NCT04850399) demonstrating the safety of intravenous administration at doses ranging from 0.75 to 20 mg in healthy volunteers. Pharmacokinetic studies have further established that AnxA5 administration at 50–100 μg/kg in patients with normal renal function results in dose-dependent plasma concentration increases, rapid clearance, and no significant impact on coagulation parameters over a 7-day observation period [50]. These data provide a strong foundation for continued clinical development of AnxA5-based therapies. Given its broad biological activities, the therapeutic potential of AnxA5 may extend beyond LoS to include a wide range of inflammatory and fibrotic conditions. We hypothesize that AnxA5 may be beneficial for treating hypertrophic scars, keloids, and radiation-induced skin injury. Preliminary, unpublished data from a bleomycin-induced pulmonary fibrosis model further support the anti-fibrotic efficacy of AnxA5 across multiple tissue types. Importantly, recombinant protein technology enables scalable production of AnxA5 with consistent quality and well-defined mechanisms of action, making it a feasible candidate for clinical application. Future studies should aim to optimize delivery platforms, establish precise dose-response relationships, and evaluate combination therapies to enhance treatment efficacy. These efforts will be critical in advancing AnxA5 as a viable therapeutic agent for fibrotic and inflammation-related diseases.

## 5. Conclusions

In this study, we provide the first evidence of the anti-fibrotic properties of AnxA5 in TGF-β-activated dermal fibroblasts. AnxA5 selectively modulates the TGF-β/Smad2 signaling pathway, leading to a significant reduction in the synthesis and secretion of key extracellular matrix components, including COLI and fibronectin. Through its combined anti-fibrotic and anti-inflammatory actions, AnxA5 effectively attenuates LoS progression under both preventative and therapeutic treatment regimens, with the preventative approach demonstrating superior efficacy. These findings support the potential of recombinant human AnxA5 as a promising candidate for clinical translation in the treatment of fibrotic diseases.

## Figures and Tables

**Figure 1 pharmaceutics-17-00986-f001:**
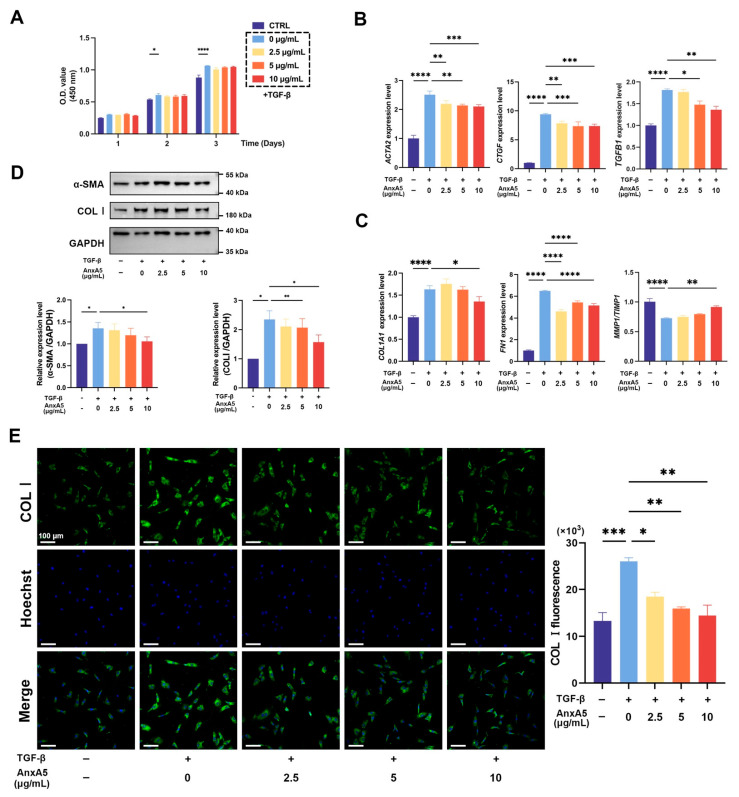
Effect of AnxA5 on TGF-β-induced dermal fibroblasts. (**A**) Cell proliferation of HFBs co-cultured with AnxA5 in a dose-dependent manner under TGF-β stimulation was detected by CCK-8 assays on days 1, 2, and 3. (**B**) The mRNA expression of ACTA2, CTGF, and TGF-β was detected by qRT-PCR in HFBs co-cultured with AnxA5 for 24 h. (**C**) The mRNA expression of COL1A1, FN1, MMP1, and TIMP1 was detected by qRT-PCR in HFBs co-cultured with AnxA5 for 24 h. (**D**) The protein expression of α-SMA and COL I was detected in HFBs co-cultured with AnxA5 for 72 h. The density values were measured relative to GAPDH. (**E**) Representative images of immunofluorescence anti-COL I staining performed in HFBs co-cultured with AnxA5 for 24 h. Quantitative analysis of fluorescence intensity of anti-COL I. Data are presented as mean ± SEM. Statistical analyses were performed by comparing to the TGF-β + 0 group, where * *p* < 0.05, ** *p* < 0.01, *** *p* < 0.001, **** *p* < 0.0001.

**Figure 2 pharmaceutics-17-00986-f002:**
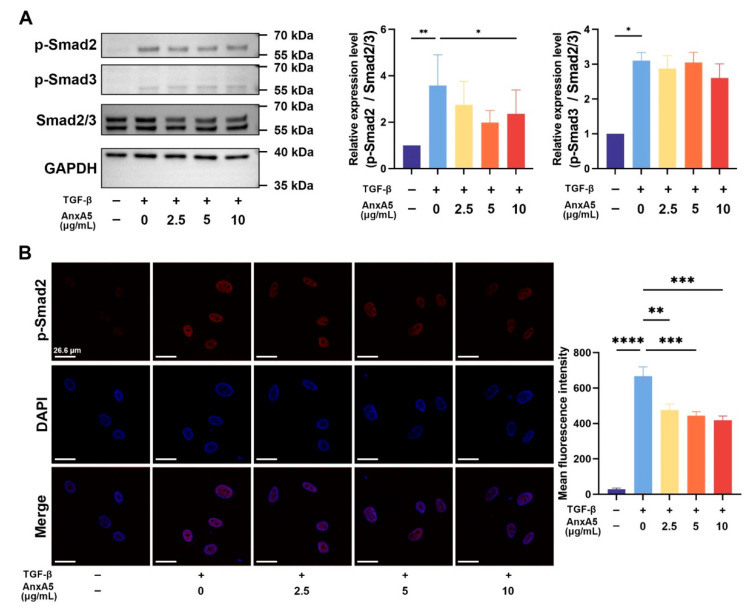
AnxA5 inhibited TGF-β/Smad2 signaling pathway. (**A**) The protein expression of p-Smad2, p-Smad3, and Smad2/3 was detected in HFBs co-cultured with AnxA5 for 1 h. Quantitative analysis of p-Smad2 and p-Smad3 protein levels relative to Smad2/3. (**B**) Representative images of immunofluorescence anti-p-Smad2 staining performed in HFBs co-cultured with AnxA5 for 1 h. Quantitative analysis of fluorescence intensity of anti-p-Smad2. Data are presented as mean ± SEM. Statistical analyses were performed by comparing to the TGF-β + 0 group, where * *p* < 0.05, ** *p* < 0.01, *** *p* < 0.001, **** *p* < 0.0001.

**Figure 3 pharmaceutics-17-00986-f003:**
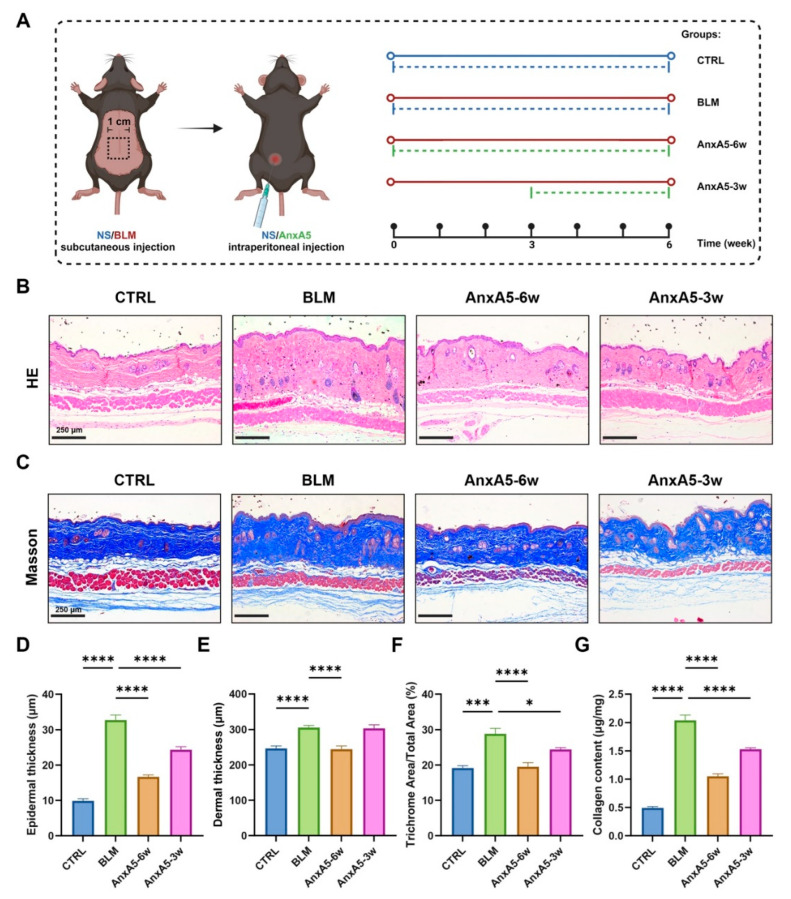
AnxA5 ameliorated bleomycin-induced LoS in murine models. (**A**) Schematic representation of the animal experiment for skin fibrosis model and treatment. (**B**) Representative images of HE staining for the skin tissue from the CTRL group, BLM group, AnxA5-6w group, and AnxA5-3w group at 6 weeks. (**C**) Representative images of Masson’s staining for the skin tissue. (**D**) Quantitative analysis of epidermal thickness (μm) of the skin tissue. (**E**) Quantitative analysis of dermal thickness (μm) of the skin tissue. (**F**) Quantitative analysis of percentage of collagen-stained area of the skin tissue through Masson’s trichrome staining. (**G**) Quantitative analysis of collagen content (μg/mg) of the skin tissue by detecting the hydroxyproline content. Data are presented as mean ± SEM, with n = 6 for each group. Statistical analyses were performed by comparing to the BLM group, where * *p* < 0.05, *** *p* < 0.001, **** *p* < 0.0001.

**Figure 4 pharmaceutics-17-00986-f004:**
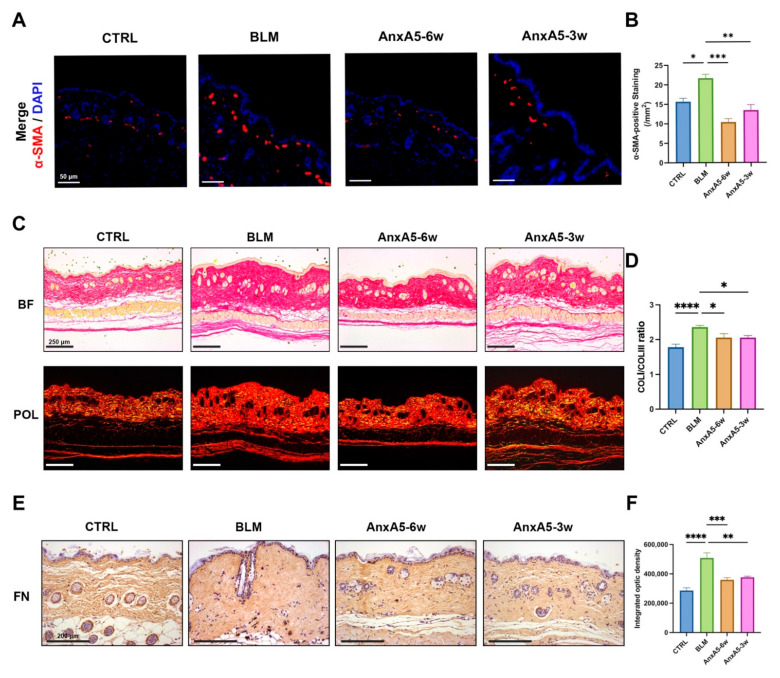
AnxA5 inhibited myofibroblast activation and ECM secretion. (**A**) Representative images of immunofluorescence anti-α-SMA staining performed for the skin tissue from the CTRL group, BLM group, AnxA5-6w group, and AnxA5-3w group at 6 weeks. (**B**) Quantitative analysis of fluorescence intensity of anti-α-SMA. (**C**) Representative images of Sirius Red staining (under bright field and polarized light device) for the skin tissue. (**D**) Quantitative analysis of COL I/COL III ratio through Sirius Red staining. (**E**) Representative images of immunohistochemical anti-FN staining performed for the skin tissue. (**F**) Quantitative analysis of integrated optic density of anti-FN. Data are presented as mean ± SEM, with n = 6 for each group. Statistical analyses were performed by comparing to the BLM group, where * *p* < 0.05, ** *p* < 0.01, *** *p* < 0.001, **** *p* < 0.0001.

**Figure 5 pharmaceutics-17-00986-f005:**
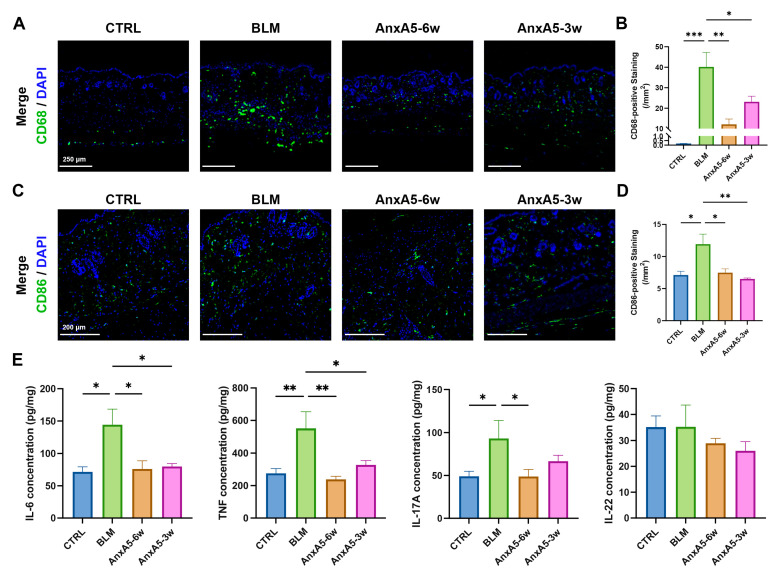
AnxA5 modulated M1-phenotype macrophage-mediated inflammation in skin fibrosis. (**A**) Representative images of immunofluorescence anti-CD68 staining performed for the skin tissue from the CTRL group, BLM group, AnxA5-6w group, and AnxA5-3w group at 6 weeks. (**B**) Quantitative analysis of fluorescence intensity of anti-CD68. (**C**) Representative images of immunofluorescence anti-CD86 staining performed for the skin tissue at 6 weeks. (**D**) Quantitative analysis of fluorescence intensity of anti-CD86. (**E**) Concentrations of TNF, IL-6, IL-17A, and IL-22 detected by ELISA in the skin tissue at 6 weeks. Data are presented as mean ± SEM, with n = 6 for each group. Statistical analyses were performed by comparing to the BLM group, where * *p* < 0.05, ** *p* < 0.01, *** *p* < 0.001.

**Figure 6 pharmaceutics-17-00986-f006:**
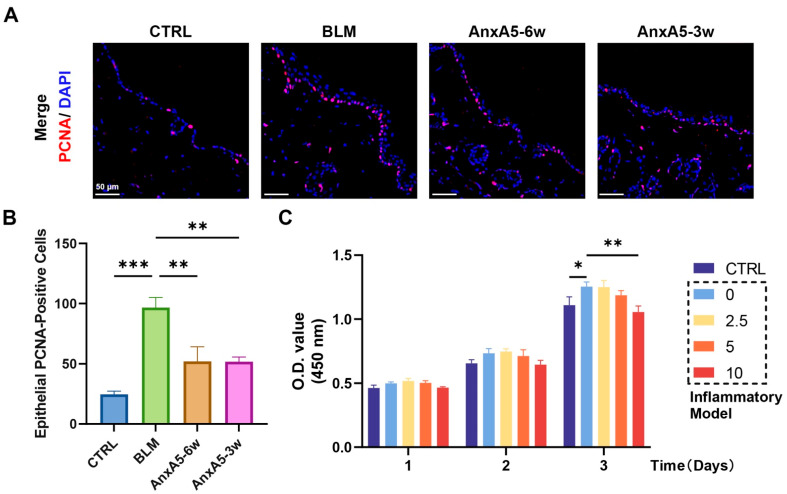
AnxA5 suppressed epidermal hyperproliferation in skin fibrosis. (**A**) Representative images of immunofluorescence anti-PCNA staining performed for the skin tissue from the CTRL group, BLM group, AnxA5-6w group, and AnxA5-3w group at 6 weeks. (**B**) Quantitative analysis of fluorescence intensity of anti-PCNA. Data are presented as mean ± SEM, with n = 6 for each group. Statistical analyses were performed by comparing to the BLM group, where ** *p* < 0.01, *** *p* < 0.001. (**C**) Cell proliferation of HaCaTs co-cultured with AnxA5 in a dose-dependent manner under inflammatory microenvironment was detected by CCK-8 assays on days 1, 2, and 3. Data are presented as mean ± SEM. Statistical analyses were performed by comparing the inflammatory model +0 μg/mL A5 group, where * *p* < 0.05, ** *p* < 0.01.

**Figure 7 pharmaceutics-17-00986-f007:**
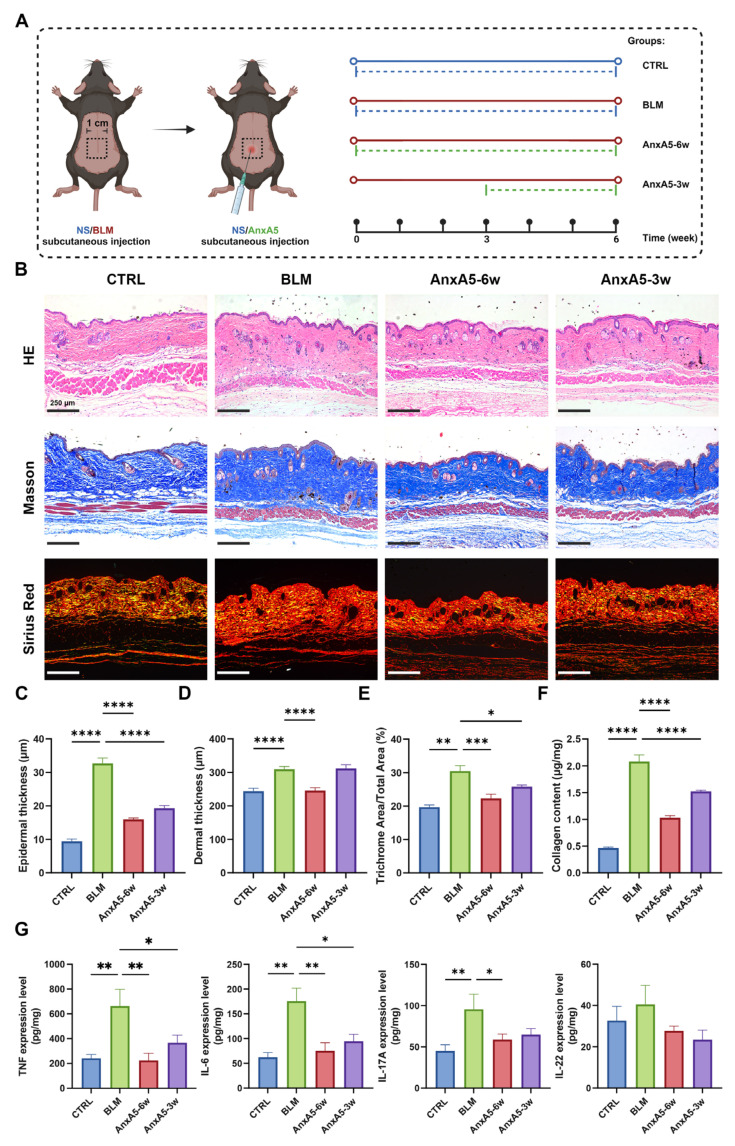
AnxA5 ameliorated BLM-induced skin fibrosis by localized treatment. (**A**) Schematic representation of the animal experiment for skin fibrosis model and localized treatment. (**B**) Representative images of HE staining, Masson’s trichrome staining, and Sirius Red staining (under bright field and polarized light device) for the skin tissue from the CTRL group, BLM group, AnxA5-6w group, and AnxA5-3w group at 6 weeks. (**C**) Quantitative analysis of epidermal thickness (μm) of the skin tissue. (**D**) Quantitative analysis of dermal thickness (μm) of the skin tissue. (**E**) Quantitative analysis of percentage of collagen-stained area of the skin tissue through Masson’s trichrome staining. (**F**) Quantitative analysis of collagen content (μg/mg) of the skin tissue by detecting the hydroxyproline content. (**G**) Concentrations of TNF, IL-6, IL-17A, and IL-22 detected by ELISA in the skin tissue at 6 weeks. Data are presented as mean ± SEM, with n = 6 for each group. Statistical analyses were performed by comparing to the BLM group, where * *p* < 0.05, ** *p* < 0.01, *** *p* < 0.001, **** *p* < 0.0001.

## Data Availability

The data and materials during the current study are available from the corresponding authors upon reasonable request.

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
