# Peer review of "Recombinant Human Annexin A5 Ameliorates Localized Scleroderma by Inhibiting the Activation of Fibroblasts and Macrophages"

_pharmaceutics, 2025, doi:10.3390/pharmaceutics17080986_

Round 1
Reviewer 1 Report
Comments and Suggestions for Authors
The manuscript by Kang et al. presents a comprehensive study focusing on the dual (anti-inflammatory and antifibrotic) role of human recombinant AnxA5 in localized scleroderma. The authors using a combination of in vitro assays and an in vivo model, demonstrate that AnxA5 reduces myofibroblasts differentiation, ECM deposition, inflammatory cytokine production and macrophages infiltration. Overall, the work is well structured and written, the experimental approach is appropriate, and the findings are novel and interesting. The major strengths of the study are the potential protein therapy with topical and systemic utility, the unraveling of the specific mechanism of action and the prophylactic and therapeutic treatment window it provides. However, a few issues should be addressed before publication.
1. Western Blot Data Representation
- the images are of low resolution without proper labeling
- some bands are oversaturated or duplicated
- lanes and marker should be clearly labeled
Please provide high resolution images with visible ladders and properly labelled. Quantify bands, normalize properly either to Gapdh or total protein, and provide full statistical analysis. If feasible, include at least three independent biological replicates.
2. The duration of TGFb treatment before AnxA5 should be referred. Was it a pre-treatment or a simultaneous treatment?
3. The study emphasizes Smad2 suppression as the sole mechanism of AnxA5 action. Were other TGFb related pathways also studied? Could you please provide the relative data? Additionally, please add this to your discussion section.
4. AnxA5's role on macrophages is limited to CD86 staining and cytokine levels in tissue lysates. You should strengthen your claims with relevant in vitro polarization experiments or else, limit interpretation to phenotypic correlation and not to mechanistic theories.
5. qPCR data for fibrotic markers lack protein-level confirmation. Please check more proteins, not only COL1A1. One or two more are needed.
6. Please include a brief discussion part explaining how the efficacy of local AnxA5 delivery could be translated to a true topical therapy in humans. How dermal delivery can be achieved? What are the challenges of using a recombinant protein in human skin?
7. Data is missing regarding the sex of animals used and the long term effect of the therapy. Even short-follow up data would be useful. Additionally, have you tried another dose or just one?
8. Minor issues/ suggestions
- figure 5 panels: in both C and D, CD68 is referred. You should correct to CD86 appropriately.
- in some figure legends, there are double commas
- sup. Fig2 can be included in the main text
Author Response
Comment 1: Western Blot Data Representation
- the images are of low resolution without proper labeling
- some bands are oversaturated or duplicated
- lanes and marker should be clearly labeled
Please provide high resolution images with visible ladders and properly labelled. Quantify bands, normalize properly either to Gapdh or total protein, and provide full statistical analysis. If feasible, include at least three independent biological replicates.
Response 1: Thank you for your suggestion. We have provided the western blot images as suggested and revised the figures. Also, we have detailed the relative method and statistical analysis in the Methods 2.4 (Page 4). The results of multiple independent biological replicates are provided at appendix 1.
Comment 2: The duration of TGFb treatment before AnxA5 should be referred. Was it a pre-treatment or a simultaneous treatment?
Response 2: Recombinant human TGF-βwas simultaneously treated with AnxA5, which revised in the Methods 2.1 (Page 3).
Comment 3: The study emphasizes Smad2 suppression as the sole mechanism of AnxA5 action. Were other TGFb related pathways also studied? Could you please provide the relative data? Additionally, please add this to your discussion section.
Response 3: Thank you for pointing out this. TGF-β signaling includes both canonical (Smad-dependent) and non-canonical (JNK, ERK, and PI3K/AKT) pathways. While these pathways exhibit cross-talk with Smad signaling and ultimately converge on nuclear Smad complexes, the canonical TGF-β/Smad pathway remains the primary mediator of fibrogenesis. In this study, we specifically detected Smad2/3 activation following AnxA5 treatment. We have added this in the Discussion (Page 16).
Comment 4: AnxA5's role on macrophages is limited to CD86 staining and cytokine levels in tissue lysates. You should strengthen your claims with relevant in vitro polarization experiments or else, limit interpretation to phenotypic correlation and not to mechanistic theories.
Response 4: The pathogenesis of localized scleroderma involves excessive inflammatory infiltration and aberrant extracellular matrix deposition, highlighting immunomodulation as a critical therapeutic target. Our in vivo investigations demonstrate AnxA5's immunoregulatory properties through histological analysis and cytokine quantitative analysis. While the current study establishes AnxA5's capacity to reduce macrophage infiltration and pro-inflammatory cytokine secretion in LoS lesions, detailed mechanistic insights into AnxA5-mediated macrophage polarization are systematically characterized in our other studies (doi:10.7150/ijbs.92802.; 10.1016/j.reth.2024.03.013.).
Comment 5: qPCR data for fibrotic markers lack protein-level confirmation. Please check more proteins, not only COL1A1. One or two more are needed.
Response 5: Thank you for your suggestion. While we conducted preliminary investigations of additional fibrotic markers, technical limitations precluded definitive analysis. Commercially available antibodies for these targets demonstrated inconsistent specificity and reproducibility under our experimental conditions. Consequently, we focused our mechanistic validation on critical, representative and well-established markers (α-SMA, COLⅠ) with rigorously validated detection reagents.
Comment 6: Please include a brief discussion part explaining how the efficacy of local AnxA5 delivery could be translated to a true topical therapy in humans. How dermal delivery can be achieved? What are the challenges of using a recombinant protein in human skin?
Response 6: We have added a brief discussion about transdermal delivery of recombinant proteins in the Discussion (Page 17).
Comment 7: Data is missing regarding the sex of animals used and the long-term effect of the therapy. Even short-follow up data would be useful. Additionally, have you tried another dose or just one?
Response 7: Thank you for pointing out this. All animal procedures utilized male C57BL/6 mice (specified in Methods 2.6) to maintain consistency with established bleomycin-induced scleroderma models. The 6-week experimental timeframe was selected based on validated protocols demonstrating complete fibrotic phase development (doi: 10.1093/bjd/ljae286.; 10.1007/s10787-018-0527-4), while the dosage of AnxA5 was based on prior studies in osteoarthritis models (doi: 10.7150/ijbs.92802). We will address long-term therapeutic efficacy and optimized dosage in the future investigations.
Comment 8: Minor issues/ suggestions
- figure 5 panels: in both C and D, CD68 is referred. You should correct to CD86 appropriately.
- in some figure legends, there are double commas
- sup. Fig2 can be included in the main text
Response 8: We have revised these suggestions in the figures and figure legends.

Reviewer 2 Report
Comments and Suggestions for Authors
- Describe the main reagent (substrate) in the CCK-8 assay kit. In section 2.2, the method was described as “cell proliferation assay”. However, in the legend to Fig. 1A, the results are presented as “cell viability”. Cell proliferation and cell viability are different terms. Please correct. In some cases, cell viability may not change, but proliferation may be enhanced or, conversely, completely inhibited. Also, please add the AnxA5 concentrations (μg/mL) in the figures.
- All figures for section 3.7 are in the Appendix. Please provide the main results for this section in the manuscript.
- There are many typos in the manuscript. Please correct them and add spaces before the references. Use the same format for section titles (all words should start with upper- or lower-case letters, see sections 2.1 and 2.2; 2.4 and 2.5).
- Please use TNF instead of TNF-α: In 1998, at the 7th International TNF Congress, TNF-β was officially renamed lymphotoxin-α and TNF-α was renamed back to TNF.
Author Response
Comment 1: Describe the main reagent (substrate) in the CCK-8 assay kit. In section 2.2, the method was described as “cell proliferation assay”. However, in the legend to Fig. 1A, the results are presented as “cell viability”. Cell proliferation and cell viability are different terms. Please correct. In some cases, cell viability may not change, but proliferation may be enhanced or, conversely, completely inhibited. Also, please add the AnxA5 concentrations (μg/mL) in the figures.
Response 1: We have detailed the reagent of CCK-8 kit in the Methods 2.2 (Page 3) and revised the figures and the figure legends as suggested.
Comment 2: All figures for section 3.7 are in the Appendix. Please provide the main results for this section in the manuscript.
Response 2: We have revised the supplementary figure 2 as Figure 7 as suggested.
Comment 3: There are many typos in the manuscript. Please correct them and add spaces before the references. Use the same format for section titles (all words should start with upper- or lower-case letters, see sections 2.1 and 2.2; 2.4 and 2.5).
Response 3: Thank you for pointing out this. We have revised the manuscript as suggested.
Comment 4: Please use TNF instead of TNF-α: In 1998, at the 7th International TNF Congress, TNF-β was officially renamed lymphotoxin-α and TNF-α was renamed back to TNF.
Response 4: We have revised the usage of TNF in the manuscript and the figures as suggested.
Round 2
Reviewer 1 Report
Comments and Suggestions for Authors
Thank you for your thorough revisions and thoughtful responses to my comments. The manuscript has been significantly improved and is close to being acceptable for publication. I appreciate the additional data, improved figure quality, and clearer methodological descriptions. However, I have a few remaining concerns and suggestions for clarification to strengthen the manuscript further:
1. Western Blot Images and Quantification
Thank you for providing revised Western blot images and additional information regarding normalization and replicates (Methods 2.4). Please ensure:
-
In Supplementary Figure 2, clearly indicate which blots correspond to each main figure panel.
2. Mechanism of Action: TGF-β Pathway Specificity
While I appreciate your focus on Smad2 signaling and the clarification added to the Discussion, I recommend adding a brief statement acknowledging that non-canonical TGF-β signaling pathways (e.g., ERK, JNK, PI3K/Akt) may also be relevant in dermal fibrosis and should be explored in future studies. This will help contextualize your findings and clarify the scope of the mechanistic investigation.
3. Macrophage Polarization
Your response appropriately cites related mechanistic studies. However, in this manuscript, macrophage modulation is inferred primarily from CD86 staining and cytokine levels. I suggest revising the language in the Discussion to clearly state that macrophage phenotype modulation was observed at the phenotypic level and that detailed mechanistic polarization data were not directly assessed in this study.
4. Fibrotic Marker Confirmation
The explanation regarding antibody limitations is acceptable. Notably, your use of α-SMA and fibronectin at the protein level (in addition to COL1) does improve the robustness of the conclusions.
5. Discussion on Translational Relevance
The added section on topical delivery is very helpful. Thank you for including discussion on microneedles, liposomal systems, and formulation challenges for recombinant proteins. This strengthens the translational perspective.
Overall, this is a strong and well-conducted study with clear therapeutic implications. With these minor clarifications and adjustments in wording, the manuscript will be well-positioned for publication.
Author Response
Comment 1. Western Blot Images and Quantification
Thank you for providing revised Western blot images and additional information regarding normalization and replicates (Methods 2.4). Please ensure:
In Supplementary Figure 2, clearly indicate which blots correspond to each main figure panel.
Response 1: We have revised the Supplementary Figure 2 as suggested.
Comment 2. Mechanism of Action: TGF-β Pathway Specificity
While I appreciate your focus on Smad2 signaling and the clarification added to the Discussion, I recommend adding a brief statement acknowledging that non-canonical TGF-β signaling pathways (e.g., ERK, JNK, PI3K/Akt) may also be relevant in dermal fibrosis and should be explored in future studies. This will help contextualize your findings and clarify the scope of the mechanistic investigation.
Response 2: We have added the statement about non-canonical TGF-β signaling pathways in the Discussion (Page 16) as suggested.
Comment 3. Macrophage Polarization
Your response appropriately cites related mechanistic studies. However, in this manuscript, macrophage modulation is inferred primarily from CD86 staining and cytokine levels. I suggest revising the language in the Discussion to clearly state that macrophage phenotype modulation was observed at the phenotypic level and that detailed mechanistic polarization data were not directly assessed in this study.
Response 3: We have revised and added a statement in the Discussion (Page 17) as suggested.
Comment 4. Fibrotic Marker Confirmation
The explanation regarding antibody limitations is acceptable. Notably, your use of α-SMA and fibronectin at the protein level (in addition to COL1) does improve the robustness of the conclusions.
Response 4: Thank you for pointing this out.
Comment 5. Discussion on Translational Relevance
The added section on topical delivery is very helpful. Thank you for including discussion on microneedles, liposomal systems, and formulation challenges for recombinant proteins. This strengthens the translational perspective.
Response 5: Thank you for your comment.